# Characterization of Polyphenolic Compounds from *Bacopa procumbens* and Their Effects on Wound-Healing Process

**DOI:** 10.3390/molecules27196521

**Published:** 2022-10-02

**Authors:** Adriana Martínez-Cuazitl, María del Consuelo Gómez-García, Oriana Hidalgo-Alegria, Olivia Medel Flores, José Alberto Núñez-Gastélum, Eduardo San Martín Martínez, Ada María Ríos-Cortés, Mario Garcia-Solis, David Guillermo Pérez-Ishiwara

**Affiliations:** 1Laboratorio de Biomedicina Molecular, ENMyH, Instituto Politécnico Nacional, Mexico City 07320, Mexico; 2Escuela Militar de Medicina, Centro Militar de Ciencias de la Salud, UDEFA-SEDENA, Mexico City 11200, Mexico; 3Departamento de Ciencias Químico Biológicas, Instituto de Ciencias Biomédicas, Universidad Autónoma de Ciudad Juárez, Ciudad Juárez 32310, Mexico; 4Centro de Investigación en Ciencia Aplicada y Tecnología Avanzada-Unidad Legaria, Instituto Politécnico Nacional, Mexico City 11500, Mexico; 5Centro de Investigación en Biotecnología Aplicada, Instituto Politécnico Nacional, Tlaxcala de Xicohténcatl 90700, Mexico; 6Departamento de Patología, Hospital General de Tláhuac, Mexico City 13250, Mexico

**Keywords:** *Bacopa procumbens* HPLC characterization, effect of polyphenolic compounds, in vitro and in vivo skin wound, collagen organization

## Abstract

Wounds represent a medical problem that contributes importantly to patient morbidity and to healthcare costs in several pathologies. In Hidalgo, Mexico, the *Bacopa procumbens* plant has been traditionally used for wound-healing care for several generations; in vitro and in vivo experiments were designed to evaluate the effects of bioactive compounds obtained from a *B. procumbens* aqueous fraction and to determine the key pathways involved in wound regeneration. Bioactive compounds were characterized by HPLC/QTOF-MS, and proliferation, migration, adhesion, and differentiation studies were conducted on NIH/3T3 fibroblasts. Polyphenolic compounds from *Bacopa procumbens (PB)* regulated proliferation and cell adhesion; enhanced migration, reducing the artificial scratch area; and modulated cell differentiation. *PB* compounds were included in a hydrogel for topical administration in a rat excision wound model. Histological, histochemical, and mechanical analyses showed that *PB* treatment accelerates wound closure in at least 48 h and reduces inflammation, increasing cell proliferation and deposition and organization of collagen at earlier times. These changes resulted in the formation of a scar with better tensile properties. Immunohistochemistry and RT-PCR molecular analyses demonstrated that treatment induces (i) overexpression of transforming growth factor beta (TGF-β) and (ii) the phosphorylation of Smad2/3 and ERK1/2, suggesting the central role of some *PB* compounds to enhance wound healing, modulating TGF-β activation.

## 1. Introduction

Wound healing is a highly ordered and synchronized process. Tissue repair progresses sequentially through three overlapping phases: inflammatory, proliferative, and remodeling. This process requires different cell types, cytokines, growth factors, and specific interactions with the extracellular matrix (ECM) [1,2,3,4].

Transforming growth factor beta (TGF-β) is a crucial cytokine during wound healing. It has widespread effects on cell growth, differentiation, migration, and deposition of extracellular matrix, inflammation regulation, and the promotion of connective tissue regeneration [5,6,7]. De-regulation of TGF-β produces abnormal wound healing, including scar formation. TGF-β stimulates Smad proteins and Smad-independent signal transducers such as mitogen-activated protein kinases (MAPK), implicating it in the pro-abnormal fibrotic effects of this factor [6,8].

New compounds such as pirfenidone (Kitoscell, CellPharma) are commercially available. It is used for the late remodeling phase of wound repair. Activities claimed by the pharmaceutical company also include the reduction of the inflammatory response, thereby controlling secondary edema and promoting a better blood supply [9]. Currently, no topically effective medication has been developed to accelerate the wound-healing process and/or prevent abnormal cicatrization. New alternatives have been sought for the treatment of wounds, and the use of pharmaceutical herbs is popular due to fewer side effects and the effectiveness of certain compounds. In Mexico, different ethnic groups in the state of Hidalgo have used *Bacopa procumbens* (Mill.) Green to treat skin wounds [10]. However, there are no scientific studies documenting the cellular and tissue effects, nor the molecular mechanism involved in wound repair. In this work, the effect of the aqueous fraction of *B. procumbens* on the proliferation, differentiation, and migration of 3T3 fibroblasts was evaluated. In addition, the process of tissue regeneration and the probable molecular effectors and signaling pathways induced by the extract in the excisional wound model of Wistar rats were studied.

## 2. Results

### 2.1. Phytochemical Screening of Bacopa procumbens Aqueous Fraction 

The profiles of the aqueous compounds were obtained by HPLC/QTOF-MS. A total of 28 compounds were identified in the *Bacopa procumbens* aqueous fraction. Based on the abundance of ions, the most representative compounds were naringenin-C-hexoside, equol 7-*O*-glucuronide, paeoniflorin, m-hydroxybenzoic acid, 4′-methoxyapigenin rutinoside, and methyl ferulate (Table 1).

### 2.2. In Vitro Assays

Proliferation. The highest fibroblast proliferation effects were displayed using 10 µg/mL and 100 µg/mL of aqueous fraction and TAE, showing after 48 h of incubation 51% and 35% of growth increment, respectively. In both cases no proliferative effect was observed after 72 h; the hexane (Hx) fraction showed 50% also using 100 µg/mL, while in contrast, using the chloroform (Chl) fraction the proliferative effect was only 25% at the same concentration (Figure 1a–d). Due to that the aqueous fraction showed the highest proliferative effect at lower concentrations, the subsequent analyses were performed using this fraction.

The proliferative effect of the aqueous fraction at 10 µg/mL was evaluated by measuring PCNA expression by Western blot assays. Results showed that the aqueous fraction induced the expression of PCNA protein at 24 and 48 h, increasing the basal expression of PCNA 60 and 110% from the control fibroblast. At 72 h the expression began to diminish (Figure 1e).

Migration. The aqueous fraction at 1, 10, and 50 µg/mL promoted fibroblast migration to an artificially made wound area, reducing the scratch area from 200 µm, at time zero, to 43.65, 73.88, and 56.55%, respectively (Figure 2a,b). At higher concentrations, as in control groups, no significant migration was observed.

Adhesion. Concentrations from 1 to 200 µg/mL of the aqueous fraction increased adhesion to fibronectin in all concentrations and incubation times used, having the best results using 1, 5, and 10 µg/mL for 30 min, increasing the cellular adhesion 100–125% in comparison to the control group (Figure 2c).

Differentiation. Expression of the α-SMA protein without or with 10 µg/mL of the aqueous fraction increased about 0.7-fold after 24 h, concerning its respective expression at 0 h. Interestingly, after 48 h expression decreased gradually in fibroblasts treated with the aqueous fraction, while in the control group, the α-SMA expression increased continuously to 1.5-fold at 48 and 72 h regarding its respective expression at 0 h (Figure 2d).

### 2.3. Macroscopic Wound Repair Effects of Polyphenolic Compounds from Bacopa procumbens

Macroscopic analysis of wound healing showed wound area reduction in the animal groups treated with polyphenolic compounds from the *Bacopa procumbens* aqueous fraction (*PB)* compared to animals in the WOT (without treatment) group or to animals treated with KC (Kitoscell®). At the beginning, the wounds were almost similar. At day 3, wounds WOT and treated with KC were larger in comparison with the wounds treated with *PB* which were smaller, presenting fine scabs (Figure 3a). On days 5 and 7, the area of wound reduction was evident in the *PB*-treated groups compared to other groups, showing moderate scabs, while in the KC group the scab was very prominent (Figure 3a). A comparison of the wound reduction area of each group (Figure 3b) showed that 50% of the wound reduction area was achieved in WOT and KC groups until days 6 and 7, respectively. Instead, the animals treated with 80 mg/mL and 160 mg/mL of *PB* achieved 50% reduction on day 5 and 4, respectively, suggesting that treatment particularly with 160 mg/mL of *PB* hydrogel accelerates wound closure by at least 48 h (Figure 3b).

### 2.4. Histological Findings of Wound Healing from Different Groups

The lesions at 3, 5, and 7 days after the injury were removed for histopathological analyses. At day 3, WOT and KC groups showed significant edema and fibrin deposits, infiltration of mononuclear cells, and a small amount of fibroblasts. On the other hand, wounds treated with both *PB* concentrations had less edema and fibrin deposits; infiltration of mononuclear and polymorphonuclear cells and the presence of rounded and disorganized fibroblasts was evident (Figure 4). In the group treated with 160 mg/mL of *PB*, some elongated fibroblasts were observed. After 5 days, wounds from all groups showed less edema; in the WOT and KC, we found some disorganized fibroblasts and an incipient formation of collagen fibers (Figure 5). However, wounds from animals treated with 80 mg/mL *PB* presented more fibroblasts (Figure 4) and an increased collagen content (Figure 4 and Figure 5); wounds from animals treated with 160 mg /mL *PB* showed an important quantity of elongated fibroblasts (Figure 4) and a better distribution and organization of collagen fibers (Figure 4 and Figure 5). On day 7, wounds from all groups showed a remarkable number of fibroblasts and a substantial reduction in inflammatory cells. In the WOT, KC, and 80 mg/mL *PB*-treated groups, the elongated fibroblasts were still disorganized, few blood vessels were formed, and the collagen fibers were moderately organized. Interestingly, in the 160 mg/mL *PB* group, we found elongated and organized fibroblasts, considerable blood vessels, and the collagen fibers were thicker and perpendicularly oriented (Figure 4 and Figure 5).

To determine the collagen types presented in the ECM of the wound repair tissues, Picrosirius Red staining was performed. Under polarized light microscopy, type III collagen, first produced by fibroblasts during wound healing, was detected by greenish birefringence, whereas mature reticular collagen type I was detected in yellow and red [11]. At day 3, both collagens were almost undetectable in the WOT and KC control groups, observing abundant interfibrillar spaces (Figure 6), while in wounds treated with both *PB* concentrations the greenish and yellow-red reticular signals were evident and disorderly arranged (Figure 6). Interestingly, wounds treated with 160 mg/mL *PB* showed thickened collagen fibers. Five days after wounding, the collagen fiber content increased in all groups. In WOT, KC, and 80 mg/mL *PB*-treated groups the fibers were thin and still disorganized, while in wounds treated with 160 mg/mL *PB*, both types of collagens increased, observing thicker and arranged fibers. After 7 days, collagen fibers tended to be better organized; in WOT the collagen fibers were still thin, whereas in the KC treatment group, the fibers tend to be arranged in the dermis layer, constituting thicker collagen fibers composed of type I and type III collagens. In wounds treated with 80 mg/mL of *PB*, collagen type I increased but the fibers, although oriented, remained thin. Interestingly, in the wounds treated with 160 mg/mL *PB*, the deposition and orientation of the collagen fibers were highly improved; the arrangement of the collagen fibers tended to be parallel to the surface of the wound and the fibers were mainly yellowish-red stained, indicating that the coarse fibers are predominantly composed by type I collagen.

Immune-histochemical analyses revealed that collagen relative expression after 3 days of the wound-healing process was similar in the four groups, representing approximately 50% of the protein expression of normal skin at day 0 (Figure 7a and Appendix A). WOT and KC-treated groups kept around 0.5 relative expression levels at 5 and 7 days, while the 80 mg/mL *PB*-treated group showed 0.6 relative expressions at the same days. Interestingly, after 5 days of the wound-healing process, 160 mg/mL *PB* treatment restored almost normal skin levels of type I collagen protein (Figure 7a). Expression of the type I collagen gene showed a differential expression pattern. At day 0, tissues from all groups showed basal collagen gene expression. At day 3, in the WOT group, collagen mRNA increased its basal expression 600 times, slightly growing up after seven days; in the KC-treated group, the collagen expression was incipient, increasing slowly, having a maximum level at day 7. Tissues from the 80 mg/mL *PB*-treated group showed the maximum level at day 5, decreasing progressively to basal expression at day 7. In contrast, the treatment with 160 mg/mL of *PB* presented an initial relative increment of 550 times, having the maximum levels after 5 days. Interestingly, at day 7, collagen relative expression dropped from 1200 to 200 times (Figure 7b).

### 2.5. Bacopa Treatment Produced Scars with Better Mechanical Properties

The mechanical properties of the scars were evaluated by tensile test after complete wound healing was achieved (24 days) in WOT, KC, and *PB*-treated groups, which showed the best wound-healing effect (160 mg/mL). The yield load represents the failure point which is directly proportional to the physical strength of the healed skin. The mechanical strength of the scar of the WOT group was 180.62 N/mm^2^. The KC-treated scar showed a mechanical strength of 222.82 N/mm^2^, while the *PB*-treated scar had a mechanical strength of 229.96 N/mm^2^, representing an increase of 23.36% and 27.31%, respectively, compared to the WOT values (Figure 7c). Picrosirius red staining showed that the deposition and alignment of collagen fibers improved with KC and *PB* treatment; however, *PB* treatment also showed greater engulfment and better collagen alignment, suggesting that the better organization of the extracellular matrix correlates with the better mechanical performance shown in *PB*-treated lesions (Figure 7d).

### 2.6. Key Molecular Effectors of Wound Repair Induced by Bacopa

The proliferative effect of the *PB* extract on wound tissue was evaluated by measuring the expression of proliferating cell nuclear antigen (PCNA) (Figure 8a and Appendix A). Results showed that PCNA expression slowly increased in the WOT group, having the maximal expression 7 days after injury. In the KC group, the PCNA expression also increased, having the expression picked at day 5, decreasing slightly at day 7. In both *PB*-treated groups, the PCNA expression was enhanced very early (day 3). Interestingly, after five days, PCNA expression declined, almost reaching the basal expression level at day 7 (Figure 8a and Appendix A).

Immunohistochemistry and RT-qPCR analyses for key molecular effectors TGF-β-, Smad2/3-p, and ERK1/2-p were conducted. TGF-β- protein expression (Figure 8b and Appendix A) reached the maximum expression levels at 7, 5, and 3 days in WOT, KC, and *PB* treatments, respectively. At day 7, the TGF-β- expression was down-regulated in the KC and 160 mg/mL *PB*-treated groups. On the other hand, analysis of TGFβ- mRNA expression showed (Figure 8c) the highest values in WOT at day 5, decreasing suddenly at day 7, while in the KC group, the TGF-β- gene was not expressed at day 3, but increased slowly at day 5, displaying four times the increment of basal expression at day 7. In contrast, in the 80 mg/mL *PB*-treated group, TGF-β- mRNA increased 1.7- and 1.6-fold at day 3 and 5, respectively, and decreased at day 7; in the 160 mg/mL *PB*-treated group, TGF-β- mRNA expression augmented 2-fold at day 3, having its maximal expression (4.5-fold) at day 5, and suddenly almost turned off at day 7 (Figure 8c).

TGF-β could act through a canonical pathway mediated by the phosphorylation of Smad2/3 [2,8,12] or by an alternative pathway that involves the MAPK, which provokes ERK1/2 phosphorylation [13]. Our results showed that Smad2/3 phosphorylated is slightly overexpressed at day 3 in the KC group and in both *PB*-treated groups. At day 5, its expression increased in all groups, and finally at day 7 its expression almost returned to the basal levels (Figure 8d and Appendix A). In contrast, the ERK1/2 phosphorylated protein remained at basal levels in WOT and KC groups at day 3, while on days 5 and 7, its expression was slightly down-regulated. Interestingly, *PB* treatment increased the phosphorylated ERK1/2 protein 1.7-fold at day 3 in both concentrations, and its expression gradually decreased at day 5 (1.4-fold), returning almost to basal levels in wounds treated with 160 mg/mL of *PB* at day 7, while in 80 mg/mL Bacopa the ERK1/2 phosphorylated protein was down-regulated, as in the WOT and KC groups (Figure 8e and Appendix A).

## 3. Discussion

It has been discovered that medicinal resources obtained from plants play an important role in the management of skin disorders [14,15]. The work presented was conducted to demonstrate the in vitro and in vivo wound-healing activity and the mechanisms and putative molecular pathways exerted by polyphenolic compounds from the *Bacopa procumbens* aqueous fraction. Our findings suggest that the 160 mg/mL polyphenolic compounds from the *Bacopa procumbens* aqueous fraction (*PB*) accelerate the wound-healing process, regulating the activation of key molecules such as TGF-β1 and ERK1/2 and the synthesis of collagen, improving the mechanical properties of the scar.

There are few reports on the chemical composition of *Bacopa procumbens*. Pathak et al., (2005) and González-Cortazar et al., 2019 reported arbutin in pure-ethanol-derived and hydroalcoholic-mixture-derived extracts, respectively. The same compound was identified in the aqueous fraction in the present study. Additionally, Gonzalez-Cortazar et al. (2019) documented the presence of shikimic acid derivatives [16,17].

The major components such as naringenin, paeoniflorin, and rutinoside have an anti-inflammatory and antioxidant effect; also, paeoniflorin regulated the apoptosis in a foot wound-healing model in diabetic rats [18,19,20]. Sharath et al., 2010, suggested that a *B. monniera* extract (BME) and a Bacoside-A compound also reduce epithelization time and increase tensile strength; however, all these components could improve only one phase [21,22,23,24,25]. An NF3 extract from *Radix astragali* and *Radix rehmanniae* improved the healing of diabetic wounds. In vitro, the NF3 extract had proliferation and angiogenic effects that reduced NO production [23].

The results by MTT and Western blot assays demonstrated that the aqueous fraction at 10 µg/mL induced fibroblast proliferation at earlier times, and this effect decreased in later times, suggesting a fine regulation of this process, considered as an important factor in dermis regeneration [24]. Similar results were found with a hydroalcoholic extract of *Plinia peruviana* at 100 µg/mL, inducing a fibroblast proliferative effect at 24 h that decreased after 48 h. However, a P. peruviana extract at 200 µg/mL had a toxic effect [25]. Instead, the aqueous fraction of *B. procumbens* at 10 µg/mL induced an increment of PCNA protein, displaying 60, 110, and 100% of proliferation at 24, 48, and 72 h compared with control at time 0. Alerico et al. in 2015 showed that an *Achyrocline satureioides* ethanolic extract using 1 µg/mL stimulated keratinocyte and fibroblast proliferation by MTT assay and by the expression of the proliferative Ki-67 at 24 h; however, the authors did not show the effect after 24 h. We found that the aqueous fraction increased proliferation in a gradual manner, and at 72 h the proliferative effect decreased, while in control fibroblasts without treatment, proliferation increased slowly, having a maximal effect at least 72 h later, suggesting that the aqueous fraction accelerated the process [26].

Cell–cell and cell–matrix interactions are relevant for the wound-healing process; in natural wound repair, key molecules such as fibronectin and collagen promote signaling for cell adhesion and cell migration, and serve as binding sites for an important number of growth factors [27,28,29]. Our results suggest that *Bacopa* improved cell adhesion at 30 min, and interestingly, after this time the adhesive effect slowly decreased. Raimoidin et al., 2000, showed that *Sedum telephium* and its polysaccharide fraction inhibited cell adhesion in fibroblasts, whereas the flavonoid fraction did not contribute to this effect [30]. On the other hand, an *Atropa belladonna L*. aqueous extract at 1% induced human dermal fibroblast, galectin, and fibronectin expression, but did not stimulate any transition from fibroblast to myofibroblast [31]. O di Martino et al., 2017, demonstrated in tissue explants that a *Hibiscus syriacus* ethanolic extract that contains flavonoids and coumarins induced fibronectin and filaggrin expression [32], suggesting that these kinds of components could be essential for the adhesion effect. We also showed, using scratch assays, that the aqueous fraction enhanced fibroblast migration at 48 h, using 1 to 50 µg/mL; a similar effect was found with a *P. cyanescens* extract [24] and with a *Stewertia koreana* ethyl acetate extract increasing the migration of human fibroblasts [33]. Talekar et al., 2017, showed that a 3 µg/mL formulation based on an aqueous extract of *Vitex negundo* L., *Emblica officinalis*, *Gaertn,* and *Tridax procumbens* L., having an important content of flavonoids, induced keratinocytes and fibroblast migration [34]. In contrast, a *P. peruviana* hydroalcoholic extract did not affect the migration rate at 24 h, although the extract had cryoprotective and antioxidant effects [25].

Skin fibroblasts differentiate to myofibroblasts by inducing the actin gene expression to acquire some contractile proprieties of smooth cells and produce wound contraction. Moreover, the participation of myofibroblasts is also related to composition, organization, and mechanical proprieties of ECM [35]. The results showed that the *B. procumbens* aqueous fraction induced myofibroblast differentiation at 24 h, drastically diminishing the effect at 72 h. These results suggest that the *B. procumbens* aqueous fraction finely regulates the wound contraction activity, an important fact regarding non-healing wounds [36]. Water et al., 2013, suggested that myofibroblasts probably sense alteration in the mechanical microenvironment and translate these into changes in gene expression to produce a proper contraction, and on the other hand, it is well known that temporally limited myofibroblast function is necessary for normal acute wound healing [37].

We demonstrated that topical treatment of wounds using 160 mg/mL of *PB* accelerated the wound reduction rate in at least 48 h. Compared to the WOT or KC groups, this effect was statistically significant in all evaluated times, suggesting that *PB* acts in inflammatory, proliferative, and remodeling phases. Süntar et al., 2013, demonstrated that the apigenin flavonoid improved wound healing after 12 days; Sharath et al., 2010, using Bacoside-A isolated from *B. monniera*, found that it improved healing after 16 days [21,38]. The results give strong evidence to suggest that *PB* has a much better wound-healing action, enhancing and accelerating the process to produce a scar with the highest quality compared to other plant extracts using a similar excisional model. In fact, our results also demonstrated that *PB* had a better effect than Kitoscell (KC), a patented drug used for wound care. While *PB* hydrogel treatment initiates its modulating action very early (three days), the wound care effects of the commercial product (KC) were observed until day seven, strongly suggesting that KC, in contrast to *PB*, functions mainly at the end of wound repair, modulating the tissue remodeling phase. The wound-healing process involves cell proliferation and extracellular matrix production, resulting in re-epithelialization [22]; we demonstrated that *PB* produces a faster and better ECM deposition in comparison to WOT or KC. During the wound-healing process, type III collagen is the first produced; it acts as a scaffolding bridge into the wound, and later type I collagen is produced to construct solid support in junction with type III [39]. We evidenced by Picrosirius red staining and by immunohistochemical methods that polyphenolic compounds from the *Bacopa procumbens* aqueous fraction improved the dynamics of collagen replacement, deposition, and arrangement. The arrangement of type I collagen fibers tended to be parallel to the surface of the wounds, similar to normal skin arrangement. Meanwhile, in the WOT or KC groups, the collagen fibrils constituted mainly by collagen type III were still aligned at random at similar experimental times as with *PB* treatments. Murthy et al., 2013, showed that after 10 days post-wound in a "wound incision model", rats treated with a *B. monniera* extract increased the hydroxyproline content, relating it to the total collagen content, but they did not demonstrate its structuration within the tissue. The correct distribution of collagen in the skin contributes to restoring skin functionality and strength of rupture [4]. We found that after 24 days of wound healing, the tensile strength within the scar increased by 27.31% in the group treated with *PB* compared to the WOT group, suggesting a better functional performance. *Acalypa indica* has been shown to have flavonoids and glycosides that accelerate wound contraction and increase tensile strength due to increased collagen deposition during day 10 post-wound [40]. Phytochemical and HPLC analyses of *PB* showed phenolic components, strongly suggesting this could be responsible for the wound repair effects.

We searched for the proliferative effects by measuring PCNA expression, and the results suggest that *PB* improved cell proliferation at earlier times, while it is down-regulated in the late phase, suggesting that the *PB* extract modulates the proliferative process in the tissue replacement, avoiding excess of dermal fibrosis and scarring, which are known to produce hypertrophic and keloid scars [41]. This reduction in the proliferative effect in the late phase could be related to the reduction in the inflammatory phase in the group treated with *PB*, correlating it with the induction of collagen synthesis at day 3.

TGF-β represents a key molecule that regulates many events around the complete wound-healing process [42,43]. It is shown that *PB* treatment stimulates an earlier expression of TGF-β1, and its expression is turned off during the resolution phase. Zhang et al., 2012, suggested by in vitro studies that an extract from *Radix astragali* and *Radix rehmanniae* induced overexpression of TGF-β1 [44]. Suh et al., in 2003, reported that some synthetic triterpenoids could mimic TGF-β1 action in macrophages, activating the Smad2 pathway [45]. Using a specific IHC assay, the results suggested that *PB* activated the TGF-β1 pathway, the canonical activating Smad2/3 complex, and ERK1/2 phosphorylation: this complex regulated cell proliferation and differentiation, cytoskeletal structure, apoptosis, and other biological reactions [16,46]. Kim et al., 2017, showed in vitro that an *S. chamaejasme* extract induced the proliferation of keratinocytes by activating ERK and Akt signaling; the extract from this plant also induced mRNA expression of collagen type I and III in vitro, suggesting that these pathways were involved in the wound-healing activity [47]. The in vivo results demonstrated that *PB* treatment regulates ERK1/2 activation in earlier phases, similar to the TGF-β1 expression, strongly suggesting that *PB* could activate the MAPK pathway, resulting in the improvement of wound healing effects; the results are in concordance with Kim’s findings, even though they only showed ERK activation in an in vitro model.

The specific interaction between polyphenolic compounds from the *Bacopa procumbens* aqueous fraction with its putative receptor(s) is currently in progress. In addition, works in progress focus on the identification and purification of the active compound(s) responsible for the wound-healing effect.

## 4. Materials and Methods

### 4.1. Preparation of Aquoethanolic Extract and Hydrogel Formulation

The *Bacopa procumbens* (Mill.) Greenm plant was harvested from the state of Hidalgo, Mexico, and identified with the voucher specimen number 1972 (Herbolaria IZTA-Flora Útil-Facultad de Estudios Superiores Iztacala-UNAM). We followed the methods of Gómez et al., 2016. The whole plant was dried and ground; 40 g of powder was extracted with aquoethanolic solution 50:50 (600 mL) at 76 °C for 4 h three times using a reflux system, and finally it was lyophilized, and the percentage yield was 30% *w/w* [48]. Total aquoethanolic extract (TAE) was obtained after removal of solvent under vacuum at 50 °C. Using 10 g of TAE, secondary extractions were carried out, resuspending it in 100 mL each of water, hexane, or chloroform for 2 h at room temperature. Then, each fraction was filtered and concentrated using a rotary evaporator under vacuum conditions to obtain aqueous (9.187 g, Aq), n-hexane (0.61 g, Hx), and chloroform (0.203 g, Chl) fractions, respectively. All dried extracts were stored at 4 °C.

The phenolic profile of aqueous fraction was characterized using chromatographic analysis coupled MS detector following the method proposed by Muñoz-Bernal et al., 2021. The HPLC system was an Agilent Infinity Series 1290 LC system with an Agilent 6500 Series Q-TOF MS (Agilent® Technologies, Santa Clara, CA, USA). A Zorbax Eclipse plus C18 column (50 mm × 2.1 mm, 1.8 μm) (Agilent® Technologies, Santa Clara, CA, USA) was used at 25 °C for the separations at a flow rate of 0.4 mL/min. The following elution program was used: 0–1 min 10 %, 1–4 min 30 %, 4–6 min 38 %, 6–8 min 60 %, 8–8.5 min 60 %, 8.5–9 min 10 %. Samples were filtered through 0.45 μm nylon filters and the injection volume was 1 μL. Conditions of Q-TOF MS were: ESI in negative mode; nitrogen as drying gas at 340 °C with a flow rate of 13 L/min; pressure of nebulizer was set at 60 psi; capillary voltage 175 V; the scanning mass to charge range of the Q-TOF mass analyzer was 100–3200 *m/z*. The identification of phenolic compounds was performed using retention times, UV/Vis spectra, and mass spectra from QTOF-MS using the Mass Hunter Qualitative software version B.07.00.

For the in vivo assay, polyphenolic compounds from *Bacopa procumbens* aqueous fraction were included in a hydrogel (water, carbopol 0.7%, glycerin 1%, hydantoin, methylchloroisothiazolinone, methylisothiazolinone 0.2%, and triethanolamine 1%) at 80 mg/mL and 160 mg/mL final concentration.

### 4.2. In Vitro Model

#### 4.2.1. Cell Culture

The mouse embryonic fibroblast cell line NIH 3T3 (3T3), obtained from the American Type Culture Collection (ATCC, Manassas, VA, USA), was grown in Dulbecco’s Minimal Essential Medium (DMEM), supplemented with 10 % heat-inactivated Fetal Bovine Serum (FBS), penicillin (100 IU/mL), and streptomycin (100 μg/mL). Cells were maintained at 37 °C in a 5 % CO_2_ humidified atmosphere and passaged after being detached from culture dishes P60 with 0.05% trypsin and 0.002% EDTA solution. Cells at 80–90% confluence were used for seeding and experiments.

#### 4.2.2. Cell Viability Assay

The 3T3 fibroblasts were incubated with TAE, Hx, Chl, or Aq for 24, 48, and 72 h, using 1, 10, 50, 100, and 200 µg/mL. MTT (3-(4,5-dimethylthiazol-2-yl)-2,5-diphenyl tetrazolium bromide) assays were used to determine cell viability. Cell proliferation was determined using crystal violet staining. The proliferation of the fibroblasts of the control group without any treatment, at each time evaluated (24, 48, or 72 h), was considered as 100% proliferation. The percentage of proliferation of each fraction at each time was compared with the respective control, according to the following Equation (1):% proliferation= 100 − [(absorbance with each treatment at 24, 48 or 72 h) × 100)/absorbance of the control at 24, 48 and 72 h, respectively](1)

#### 4.2.3. In Vitro Scratch Wound-Healing Assay

The 3T3 fibroblasts were seeded in 6-well plates and cultured to 90-95% confluence, using the culture conditions described above; the medium was replaced with mitomycin-supplemented medium (10 µg/mL) (MITOLEM, Leremy, CDMX, Mexico) for 2 h. The scratch wound assay was performed as previously described by Rodriguez et al., 2005 [49]. Briefly, in the middle of the cell monolayer, a scratch was made with a P200 pipette tip to mimic a wound, and cell debris was removed by two washes with PBS. The cultures were then incubated with fresh DMEM medium (2% FBS) and with the different concentrations of the aqueous fraction (1, 10, 50, 100, and 200 µg/mL) for 48 h. As controls, fibroblasts were cultured with medium supplemented with 2% SFB without mitomycin and with 2% SFB and 20 µL water (V). Cells were photographed on an inverted microscope with a 40× objective and densitometric analysis was performed using Image J version1.4.7 software (WayneRasband, National Institutes of Health, USA). Ten different fields (each 1.6 × 1.6 mm contiguous to the wound edge) were analyzed for each sample tested, and area between cells were averaged.

#### 4.2.4. Cell Adhesion Assay

Ninety-six-well plates were coated with serum-purified fibronectin (10 mg/mL), incubated overnight at 4 °C for 12 h, and blocked with 1% bovine serum albumin (BSA) in phosphate-buffered saline (PBS) at 4 °C for 2 h. Fibroblasts (104 cells/well) were incubated with different concentrations (1, 5, 10, 100, and 200 µg/mL) of aqueous fraction during 30, 60, 120, and 180 min. Adhered fibroblasts were fixed with paraformaldehyde (4%) and stained with 0.1% crystal violet (Sigma-Aldrich, Munich, Germany).

#### 4.2.5. Western Blotting Analysis

Fibroblasts were treated with 50 µg/mL of the aqueous fraction for 24, 48, and 72 h, in 2% FBS fresh medium. The cells were resuspended in ice-cold lysis buffer (10 mM/L Tris-HCL, 5 mM EDTA, 150 mM/L NaCl, 0.1 % SDS, 1% NP-40, 1% sodium deoxycholate) containing a protease inhibitor cocktail (1 mml/L phenylmethyl sulfonyl fluoride, 100 µg/L leupeptin, and 2 µg/L aprotinin). The protein concentrations were determined using Bradford protein quantification assay (Biorad, Hercules, CA, USA). Protein lysates (30 µg) were subjected to electrophoresis on 10% SDS-polyacrylamide gels and transferred to 0.45 µm nitrocellulose membranes. The membranes were stained with 0.2 % Ponceau S red to ensure equal protein loading. After blocking with 5% milk in PBS-Tween, membranes were incubated with primary rabbit antibodies against α-SMA (1:200, Santa Cruz Biotechnology, Inc, Dallas, TX, USA), PCNA (1:200, Santa Cruz Biotechnology, Inc, Dallas, TX, USA), or β- actin (1:1500, Santa Cruz Biotechnology, Inc, Dallas, TX, USA) for 2 h. Finally, the membranes were washed three times, 10 min each, and subsequently placed in the development buffer (30% H_2_O_2_, and 0.5 mg/mL 3′3-Diaminobenzidine).

All in vitro experiments were performed in triplicate.

### 4.3. In Vivo Skin Wound Model

#### 4.3.1. Animals

Male 220–240 g Wistar rats were maintained at 26 °C under 12:12 h light/dark cycle. The animal received chow and water ad libitum. This experimental work followed the guidelines of the Norma Oficial Mexicana Guide for the use and care of laboratory animals (NOM-062-ZOO-1999) and the disposal of biological residues (NOM-087-ECOL-1995). The ethics committee of ENMyH postgraduate section approved the experimental procedure of this study (approval number CBE009/2019).

The animals were divided into four randomized groups: untreated group (WOT), control group using Kitoscell® (KC), CellPharma, CDMX, Mexico [9], and two experimental treated groups using 80 mg/mL and 160 mg/mL *PB*. The animals were anesthetized, the dorsal region was chemically shaved (Veet cream, Reckitt Benckiser, Slough, UK) and harvested, and four complete thickness excision surgeries of 1 cm^2^ were performed [50]. The wounds were treated topically each day with 100 µL per wound of 8% KC or with 80 and 160 mg/mL of *PB*. The healings were observed and photographed by a digital camera 16 MPX at 15 cm away using a pedestal. Immediately after surgery and at 3, 5, and 7 days, measurements were performed using a Vernier caliper; after photographed digitization, the wound area was measured using Image Pro plus software (Media Cybernatics, Inc., Rockville, MD, USA) to calculate the percent of wound reduction, then the formula (Equation (2)):% of wound reduction = 100 − [(final area × 100)/initial area] (2)
was applied. At 3, 5, and 7 days, wound lesions were removed with adjacent healthy skin, fixed in 4% buffered paraformaldehyde, embedded in paraffin and 5 μm thickness sectioned, and then were stained with hematoxylin and eosin, Masson’s trichome, and Picrosirius Red reagent (Abcam, Cambridge, UK), using protocols described on our previous work [51,52,53]. Finally, tissue samples from the six rats of each experimental group were examined and analyzed in three different areas from three consecutive sections and photographed using the Olympus DP74 system (Olympus, Tokyo, Japan) or using a polarizing light microscope (Nikon, Tokyo, Japan).

For tensile test, WOT, KC, and *PB* at 160 mg/mL animal groups were sacrificed at 24 days, and the wound-healing skin tissue was cut with a dumbbell shape using a custom-made metal template. The tensile test was performed using the Texture Analyzer ta-xt2/1 (Stable Microsystems, Vienna Cour, England) using a 100 N load cell at a constant strain rate of 1 mm/sec.

#### 4.3.2. Immunohistochemistry

Protein detection was performed using the PolyDetector HRP/DAB Mouse/Rabbit Detection System (Bio SB, Santa Barbara, CA, USA). Tissues were rinsed with PBS and epitope retrieval was performed in a pressure cooker using the Immune Retrivever Citrate Solution (Bio SB, USA). Slides were cooled to room temperature and incubated in Peroxidase Block quenching buffer (Bio SB, USA) for 20 min to block endogenous peroxidase activity. After sections were washed with PBS, nonspecific binding sites were blocked for 30 min with BSA, Cohn fraction V, pH 7.0. Then, sections were incubated 16–18 h at 4 °C with primary antibodies: mouse monoclonal anti-collagen type I (Calbiochem, San Diego, CA, USA); PCNA (Santa Cruz Biotechnology, Dallas, TX, USA); TGF-β1 (Santa Cruz Biotechnology, USA); Smad2/3-p (Santa Cruz Biotechnology, USA); and ERK1/2-p (Santa Cruz Biotechnology, USA), all diluted 1:100. Following three washes with PBS, samples were incubated for 30 min with secondary antibody (PolyDetector HRP label; Bio SB, Goleta, CA, USA), counterstained with hematoxylin, and mounted in GVA-mount reagent (Zymed, San Francisco, CA, USA). Negative controls without primary antibody were carried out. The slides from each experimental group from three consecutive sections were observed and photographed using the Olympus system (Appendix A). Quantification of expression protein was performed by pixel measure using the Image-Pro Premier software (Media Cybernatics, Inc., Rockville, MD, USA).

#### 4.3.3. Quantitative Real-Time PCR

Total RNA was extracted from normal and wound tissues using Trizol reagent according to the manufacturer’s instructions (Invitrogen, USA). For reverse transcription, SuperScript II Reverse Transcriptase kit (Invitrogen, USA) was used to obtain cDNAs for qRT-PCR assays. Primers used for qRT-PCR were collagen type I 900 nM forward 5′-CAACCTGGATGCCATCAAGG-3′ and reverse 300 nM 5′-ATCGGTCATG-CTCTCTCCAAA-3′; TGF-β1 900 nM forward 5′-GCAGTGGCTGAACCAAGGAG-3′ and 300 nM reverse 5′-TCGGTTCATGTC-ATGGATGG-3′; GADPH 50 nM forward 5′-CACCACCAACTGCTTAGCCC-3′ and reverse 50 nM 5′-TCTGAGTGGCAGTGATGGCA-3′. All reactions were performed using SYBR Green PCR master mix (Applied Biosystems, Waltham, MA, USA) in 7300 Real-Time PCR System (Applied Biosystems, Waltham, MA, USA). As an internal control, glyceraldehyde 3-phosphate-dehydrogenase (GAPDH) was used in parallel to the target genes.

By monitoring the real-time increase in fluorescence using the SYBR Green PCR Master Mix (Applied Biosystems), the relative quantification was calculated using the CT method, which uses the formula 2^−ΔΔCT^ [54]. Relative RNA levels of target genes were normalized against the housekeeping gene GADPH. Statistically significant differences in gene expression between non-treated (0) and at different times (3, 5, and 7 days) of WOT and treatment with KC and 80 and 160 mg/mL of *PB* were analyzed by comparisons of the means of three independent biological replicates in triplicate using the statistical analysis described in Section 4.4.

### 4.4. Statistical Analysis

Statistical significance was analyzed using Two-Way ANOVA and post hoc Tukey test. All analyses were performed using Graph Pad Prism software version 7.0.

## 5. Conclusions

All the results presented suggest that the 160 mg/mL polyphenolic compounds from the *Bacopa procumbens* aqueous fraction improve and accelerate the wound-healing process. Regulating more than one event of different wound phase processes, this treatment reduces the inflammatory phase and modulates TGF-β1 action, promoting a proliferative effect and also enhancing collagen synthesis and correct deposition, resulting in optimal physiological and mechanical proprieties of scar tissue repair.

## Figures and Tables

**Figure 1 molecules-27-06521-f001:**
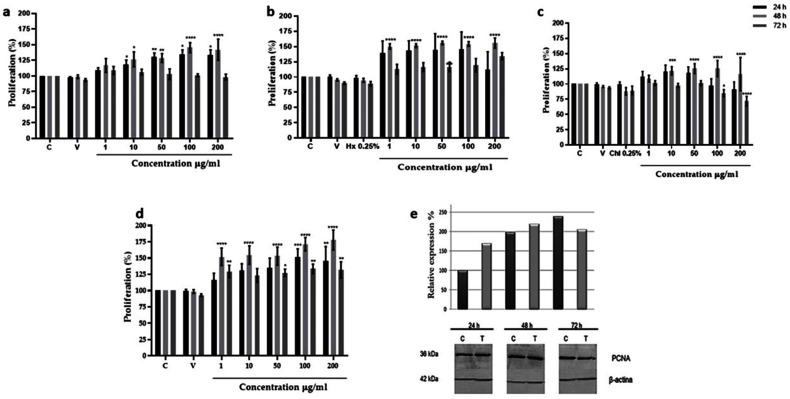
Proliferative effect of *Bacopa procumbens* extract and fractions on 3T3 fibroblasts. (**a**) Fibroblasts grown in 2% SFB-supplemented media were treated with TAE, (**b**) Hx, (**c**) Chl, or (**d**) aqueous fractions at 1, 10, 50, 100, and 200 µg/mL for 24, 48, and 72 h. Cell proliferation was measured by MTT. Control was only 2% SFB, as solvent vehicle (V), fibroblasts were grown with water (for TAE and aqueous fraction) or DMSO (for Hx and Chl), 0.25% hexane (Hx), 0.25% chloroform (Chl), or without any solvent (C). (**e**) Total protein extracts from 3T3 fibroblast treated with 10 µg/mL of aqueous fraction were extracted, and WB was carried out to analyze PCNA expression using anti-PCNA antibodies. As for loading control, anti-β actin antibodies were used. The graph shows the densitometric analysis of PCNA expression, protein extracts from 3T3 fibroblasts without treatment (C), and protein extracts from 3T3 fibroblast treated with 10 µg/mL of an aqueous fraction (T); * *p* < 0.05, ** *p* < 0.005, *** *p* < 0.001, **** *p* < 0.0001 versus control. Two-Way ANOVA, post hoc Tukey´s test. The results are expressed as means and bars SD.

**Figure 2 molecules-27-06521-f002:**
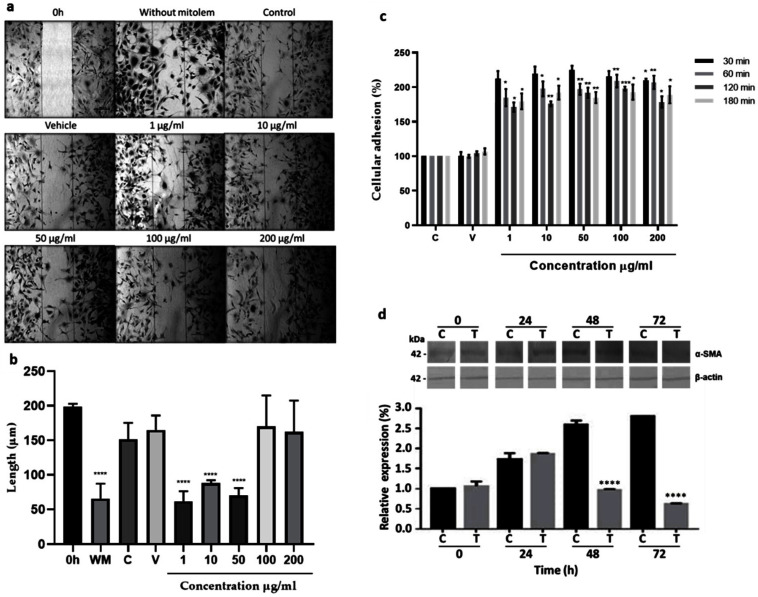
In vitro effects on 3T3 fibroblast. (**a**) Migration. Cells were grown at 80% confluence, and scratch was conducted (0 h). The 3T3 fibroblasts were grown with 2% SFB supplemented media without mitomycin (WM), with 10 µg/mL of mitomycin (control, C) to inhibit cell proliferation, with 10 µg/mL of mitomycin and water (vehicle, V), or with mitomycin and different aqueous fraction concentrations (1, 10, 50, 100, and 200 µg/mL) for 48 h. (**b**) The measure of the wound area, **** *p* < 0.0001 versus time 0. (**c**) Adhesion effect. Culture flasks were covered with fibronectin as described in methods. Cells were grown with 2% SFB supplemented media (C) and treated with 1, 10, 50, 100, and 200 µg/mL of aqueous fraction at 30, 60, 120, and 180 min. Fibroblasts grown with 2% SFB-supplemented media (C) or with water (V) are shown; * *p* < 0.05, ** *p* < 0.005, *** *p* < 0.0005, **** *p* < 0.0005 versus control. Two-Way ANOVA, post hoc Tukey´s test. (**d**) Differentiation to myofibroblast. Fibroblasts were treated with 2% SFB supplemented media or with 10 µg/mL of aqueous fraction (T) for 24, 48, and 72 h. Total protein extracts were prepared and analyzed by WB using a specific anti-α-SMA antibody for myofibroblast identification. As a loading control, anti-β actin antibody was used. The graph shows the densitometric analysis of α-SMA expression. Results are expressed as means and bar is SD. Two-Way ANOVA, post hoc Tukey’s test.

**Figure 3 molecules-27-06521-f003:**
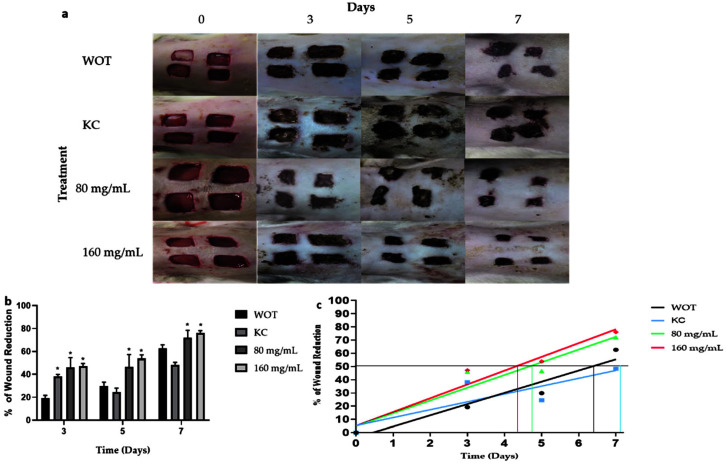
Macroscopic changes induced by *PB* during wound healing. (**a**) Representative photographs of wounds during 7 days of treatment. WOT, without treatment; KC, Kitoscell; and 80, 160 mg/mL of *PB*. (**b**) Percentage reduction in wound area with the different treatments. (**c**) Linear regression of % wound reduction in each group, the horizontal line represents the 50% wound reduction, the intersection with the lines of the experimental groups indicates the times required to reach 50% wound reduction. Values are expressed as mean ± SEM of 6 animals per group; * *p* < 0.05 vs. WOT as indicated by analysis of variance (ANOVA) and Tukey’s post hoc test.

**Figure 4 molecules-27-06521-f004:**
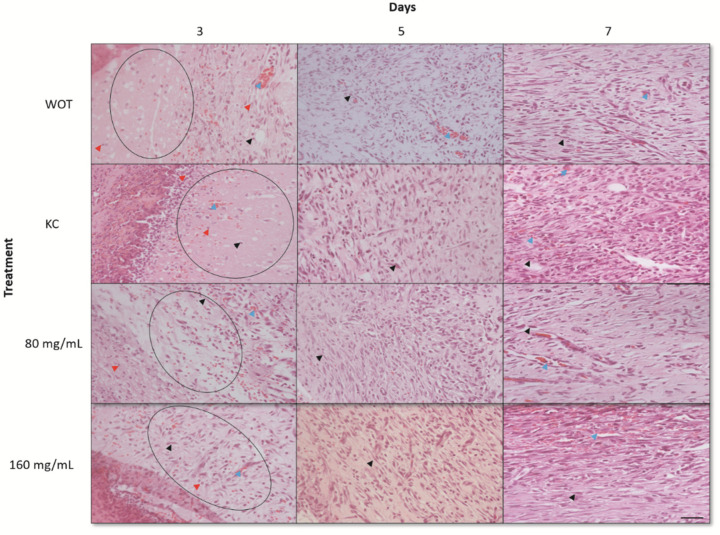
Hematoxylin and eosin (H&E) staining of wounds. Representative microphotographs at 3, 5, and 7 days post-wounding in WOT, KC, and 80 and 160 mg/mL of *PB*. Edema tissue (open circles), inflammatory cells (red arrow), fibroblasts (black arrow), blood vessels (blue arrow). Bar = 20 µm.

**Figure 5 molecules-27-06521-f005:**
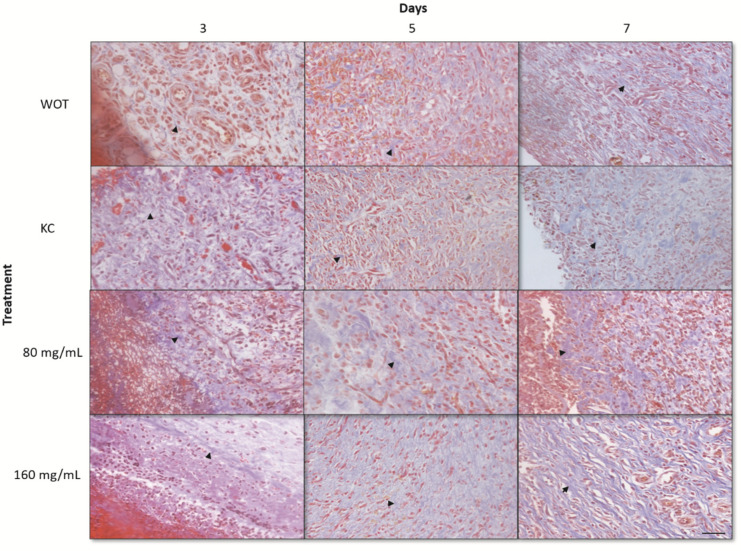
Masson’s trichrome staining of wounds. Representative microphotographs at 3, 5, and 7 days post-wounding in WOT, KC, and 80 and 160 mg/mL of *PB*. Collagen fibers (black arrow). Bar = 20 µm.

**Figure 6 molecules-27-06521-f006:**
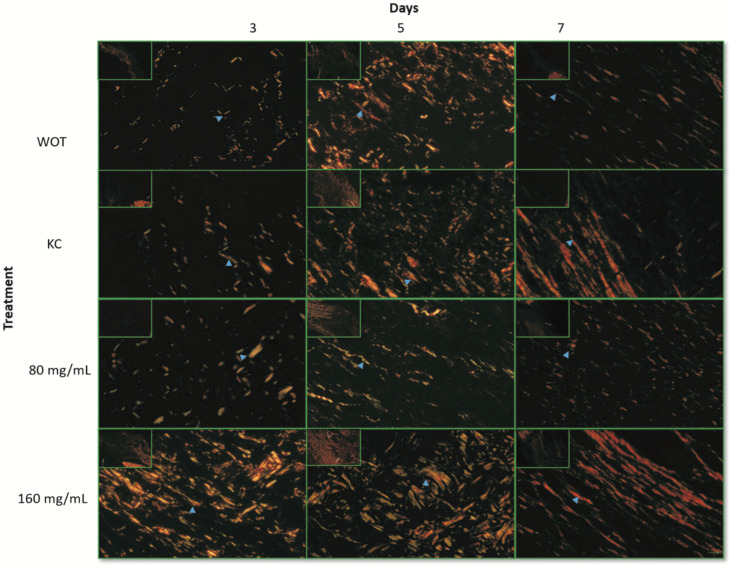
Collagen expression patterns through wound repair process. Representative microphotographs of Picrosirius Red staining under polarized light. Greenish fibers (collagen type III) and yellow-red fibers (collagen type I). Collagen fibers (blue arrow). At 3, 5, and 7 days post-wounding in WOT, KC, and 80 and 160 mg/mL of *PB*. Bar = 10 µm.

**Figure 7 molecules-27-06521-f007:**
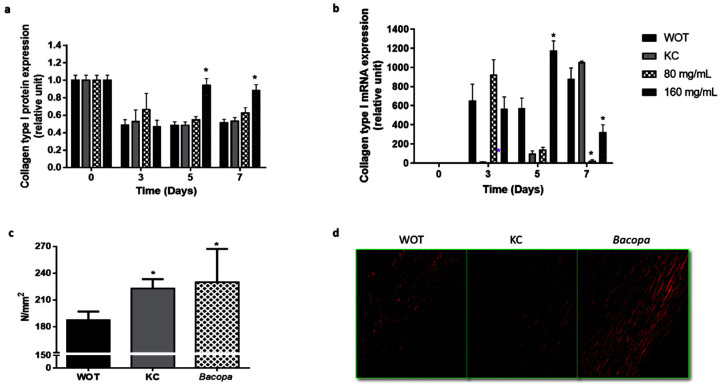
Collagen type I expression and mechanical effect and organization. (**a**) Collagen type I from wound tissues was analyzed by immunohistochemistry and by (**b**) qRT-PCR at different periods of time (0, 3, 5, and 7 days). (**c**) The tensile load applied in wounds was analyzed in scars from WOT, KC, and 160 mg/mL *PB*-treated groups after 24 days and (**d**) representative microphotographs of Picrosirius Red stain in scars from WOT, KC, and *PB* groups after 24 days of wound healing. Values are expressed as mean ± SEM of at least 6 animals per group; * *p* < 0.05 versus WOT as indicated by analysis of variance (ANOVA) and post hoc Tukey’s test.

**Figure 8 molecules-27-06521-f008:**
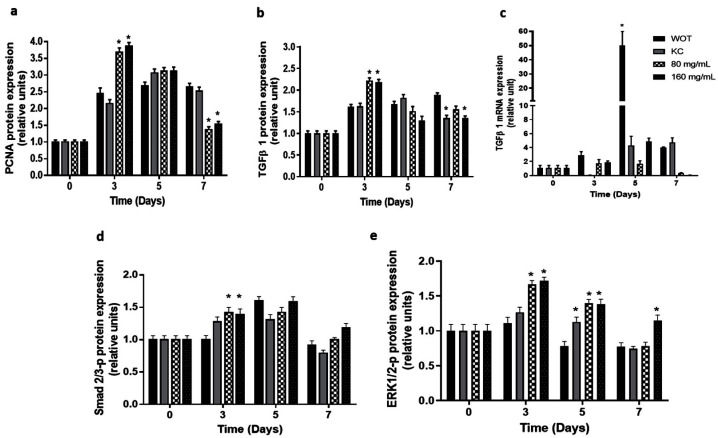
Expression of key regulators of skin wound healing from canonical and non-canonical pathways. (**a**) Proliferating cell nuclear antigen (PCNA) in wound tissue was analyzed by immunohistochemistry. Transforming growth factor beta (TFGβ1) analyzed by (**b**) immunohistochemistry and (**c**) qRT-PCR, respectively, in WOT, KC, and 80 and 160 mg/mL *PB*-treated groups at different periods of time (0, 3, 5, and 7 days). Phosphorylated (**d**) Smad2/3 and (**e**) ERK1/2 analyzed by immunohistochemistry in WOT, KC, and 80 and 160 mg/mL Bacopa-treated groups at different periods of time (0, 3, 5, and 7 days). Values are expressed as mean ± SEM of at least 6 animals per group; * *p* < 0.05 versus WOT as indicated by analysis of variance (ANOVA) and post hoc Tukey’s test.

**Table 1 molecules-27-06521-t001:** Phenolic profile of aqueous fraction of *B. procumbens*.

Compound	RT (min)	[M−H]^–^	Reference Mass
Arbutin	0.386	271.0821	272.0896
o-Hydroxybenzoic acid	0.924	137.0242	138.0317
m-Hydroxybenzoic acid	0.959	137.0245	138.0317
Feruloyl glucose	1.161	355.1033	356.1107
Esculetin	1.263	177.0188	178.0266
3-*O*-Caffeoyl shikimic acid	1.937	335.0771	336.0845
Catalposide	2.88	481.134	482.1424
Homovanillyl alcohol	3.082	167.0711	168.0786
Genistein	3.285	269.0452	270.0528
p-Coumaric acid	3.318	163.0396	164.0473
Z-Astringin	3.486	405.118	406.1264
Naringenin-C-hexoside	3.588	433.1126	434.1213
Astilbin	3.823	449.1092	450.1162
m-Coumaric acid	3.857	163.0398	164.0473
Equol 7-O-glucuronide	4.092	417.1183	418.1264
Syringaresinol-glucoside	4.093	579.2096	580.2156
p-Hydroxybenzoic acid	4.194	137.0243	138.0317
Daidzein 7-O-glucuronide	4.531	429.084	430.09
4′-Methoxyapigenin rutinoside	5.069	591.1709	592.1792
Prunetin	5.071	283.0602	284.0685
Paeoniflorin	5.104	479.1539	480.1632
Orobol	5.204	285.0409	286.0477
Phloretic acid	5.339	165.0551	166.063
Methyl ferulate	5.34	207.0658	208.0736
Urolithin C	5.745	243.0296	244.0372
Koparin	5.912	299.0551	300.0634
Coumarin	6.115	145.0298	146.0368
Stevenin	6.822	283.0606	284.0685

## Data Availability

The data that support the findings of this study are available on request from the corresponding author.

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
