# Peer review of "Characterization of Polyphenolic Compounds from Bacopa procumbens and Their Effects on Wound-Healing Process"

_molecules, 2022, doi:10.3390/molecules27196521_

Round 1
Reviewer 1 Report
The article deals with an important and interesting topic of searching for new compounds of plant origin that accelerate wound healing.
After reading the manuscript, I would like to kindly ask you for more details or to revise certain elements of the manuscript.
1. The work contains punctuation and editorial errors, eg lines 443-447: repeated fragment. Line 502: "perfomed12" and lines 515-516: "protocols13" - this should probably be a reference to one of your references. There is no explanation of the abbreviations used, eg "WM" in Figure 2b. There is no explanation of the symbols used, eg the symbol "&" in Figure 2b. On the X-axis, there is the letter "C" twice - Figure 1b.
2. What was the repetition rate for all experiments except animal studies?
3. What data are in Figure 1 (a, b, c, d, e) and figure 2 (b, c, d) - are they means or medians?
4. The symbol "**" typically signifies a significance level of p <0.01. It is wrong to use this symbol to denote a different kind of comparison - in this case, an assessment of changes over time.
5. Figure 1d: Were the same analyzes performed for the 200 µg / ml dose?
6. Lines 99-102: Looking at the graph, it seems that the best result was obtained for the dose of 1 µg / ml, while the text states that the best effect was obtained for the dose of 10 µg / ml (73.88%).
7. Figure 1 a, b, c, d: Do the graphs show the results of the cell proliferation analysis as a % of control? This is not stated in the description or in the Y-axis legend.
8. Line 108: none of the bars exceeds 2 after 24h. How do you know that the increase is significant since there are no results of the statistical analysis?
It is important to note here that despite the increase in the expression level (still no result of the statistical analysis of changes over time) there is probably no significant difference in the expression level between C and T after 24h.
9. Figures 3,7,8: Which group was the control group in the statistical analyzes?
10. Figure 3b: it is completely unreadable. A shift should be applied for each of the categories (WOT, KC, 80 mg/ml, 160 mg/ml).
11. Line 229: How was it calculated that the increase was 18.76% and 22.56% when the two groups of PB (160mg /ml) and WOT are compared?
12. Lines: 229-231: How has this correlation been proven?
13. The line charts in supplementary are unmarked, ie it is not known which line relates to which group.
14. Figure 8: why the mRNA expression level for PCNA, Smad 2/3-p, and ERK 1/2-p was not determined?
15. Lines 515-516: What does "standard protocol" mean? Is it a standard protocol for the authors of a given work or for the majority of researchers in the world? The quoted work is written in a language other than English.
16. Lines 520-524: why such studies were not carried out at the dose of 80 mg/ml?
Reviewer 2 Report
In this manuscript, authors describe the role of some Polyphenolic compounds from Bacopa procumbens to enhance wound healing, modulating TGF-β activation. They discuss data from both in vitro and in vivo experiments, using an Aqueous fraction of the bioactive compounds of the plant for the in vitro experiments and a PB hydrogel for the in vivo data.
I have many concerns:
1- English and typing must be revised in some points (as example…443-447 is the same as 425-429; lines60-64, not clear; line 395, that; lines 549-555, some special characters……)
2- Methods:
- the extraction procedure of PB is not clear. in 4.1, Authors talk about “… aquoethanolic solution to extract all the bioactive compounds (TAE). Then they chose the Aqueous fraction to make their experiments. Table 1 represents the phenolic compounds in TAE, but it is not described in the paper which of them are really present in the different fraction (Aqueous, hexane or chloroform). This is an important point also related to the different solubility of phenolic compounds to the different solvents
-related to this: why in the “vehicle group” they use DMSO (line 91)? This is the vehicle of what in the treated group?
- Results and Method: when data are calculated as %, please provide a definition of the meaning of the percentage, how it is calculated…
- as before, in Realtime data (figure 7, panel b; Figure 8, panel c), how has the relative expression been calculated? Which is the reference?
3- I’m puzzled about the assessment of cell proliferation, which is significantly reduced after 72 hours of treatment, independently from the doses of the treatment; it’s the same for PCNA expression. Authors suggest that the earlier increase in fibroblast proliferation followed by the decrease represents a fine regulation of this process. I have some doubt about the toxicity of the treatment after 72 hours: this because, if true, this would be a very good and precise regulatory system by these compounds in order to tune the healing process, even quicker than the inflammatory process itself that requires a bit more time to run out. Experiments to demonstrate that the number of cells was not influenced by cell death or by apoptosis should be added.
4- Figure 2: not clear. Panel a): non clear the different biological groups, mostly which photos are with or without mitomycin. Panel b): how the measures of the wound area were made? A precise description of how many photos were done for each point, of how measures will be calculated and analyze should be present in the test. Panel c) and d): data are expressed as %...but of what? Please clarify, also in methods. Panel d): the WB panel is poorly convincing; it's my opinion that is not original rather that due to rendering interpretation. Please provide original blot.
5- Discussion: it should be revised after the adjustments of methods, figures and results
Round 2
Reviewer 1 Report
The authors answered all questions and made many important changes to the article that increase its quality and scientific value.
Please respond to the comments below or correct a few passages:
1. Lines 101-102: specify which scatter values are shown on the graph (is it SD, SEM, IQR?).
2. Figure 2d: Was a statistical analysis of these results carried out?
3. Lines 571-573: What kind of ANOVA was used in the statistical analyzes?
Reviewer 2 Report
Dear Authors, I have just two minor points that do not convince me completely( I rewrite here the points and my comments)
first:
My first Comment. as before, in Realtime data (figure 7, panel b; Figure 8, panel c), how has the relative expression been calculated? Which is the reference?
Authors' Answer: For real time analysis (mRNA expression): Normalization and fold changes were calculated using the ΔΔCt method using the Livak formula [1], as an internal control, glyceraldehyde 3-phosphate-dehydrogenase (GAPDH) in parallel to the target genes. (Please see lines 568-570)
1. K. J. Livak, T. D. Schmittgen., “Analysis of relative gene expression data using real-time quantitative PCR and the 2(-Delta Delta C(T)) Method” Methods, 25, 4, 402-8 (2001)
My new Comment: As indicated by Livak et al., paragraph “1.3. Selection of Internal Control and Calibrator for the 2-DDCt Method”, standard housekeeping genes as GAPDH usually suffice as internal control genes... Using the 2-DDCt method, the data are presented as the fold change in gene expression normalized to an endogenous gene and relative to the untreated control...Situations exist where one may not compare the change in gene expression related to an untreated control…”
What I mean is that the 2-DDCt Method is a “Relative quantification Method”. Reference/calibrator needs to be indicated for any experiment calculation if the 2-DDCt Method is used… in the paper, untreated controls, different time points, different treatments, could have been used as reference …(see Livak et al., paragraph 1.3).
Comment 3: I am puzzled by the assessment of cell proliferation, which is significantly reduced after 72 hours of treatment, ....
Authors' Answer: Thanks for the observation, the confusion comes from the fact that it was not specified neither in the text nor in the legend of the figure that the percentages of proliferation of each of the times (24 48 and 72 h), of each of the experimental groups, were normalized with the corresponding times of the proliferation control (C), which did not receive any treatment. It is important to mention that at 72 h the nutritional and space resources in the culture decrease, therefore, in addition to the higher proliferation observed at the 24 and 48 h times in the experimental groups, it is very likely that cell proliferation at 72 h is also limited by these culture growth variables. However, as you can see in the graph below corresponding to the aqueous fraction, the proliferation curves at the different times and concentrations have a similar growth pattern to the controls, indicating that there is no toxic effect of the compounds@font-face {font-family:"Cambria Math"; panose-1:2 4 5 3 5 4 6 3 2 4; mso-font-charset:0; mso-generic-font-family:roman; mso-font-pitch:variable; mso-font-signature:3 0 0 0 1 0;}@font-face {font-family:Calibri; panose-1:2 15 5 2 2 2 4 3 2 4; mso-font-charset:0; mso-generic-font-family:swiss; mso-font-pitch:variable; mso-font-signature:-536859905 -1073732485 9 0 511 0;}p.MsoNormal, li.MsoNormal, div.MsoNormal {mso-style-unhide:no; mso-style-qformat:yes; mso-style-parent:""; margin:0cm; mso-pagination:widow-orphan; font-size:11.0pt; mso-bidi-font-size:12.0pt; font-family:"Calibri",sans-serif; mso-ascii-font-family:Calibri; mso-ascii-theme-font:minor-latin; mso-fareast-font-family:"Times New Roman"; mso-fareast-theme-font:minor-fareast; mso-hansi-font-family:Calibri; mso-hansi-theme-font:minor-latin; mso-bidi-font-family:"Times New Roman"; mso-bidi-theme-font:minor-bidi; mso-fareast-language:EN-US;}.MsoChpDefault {mso-style-type:export-only; mso-default-props:yes; font-family:"Calibri",sans-serif; mso-ascii-font-family:Calibri; mso-ascii-theme-font:minor-latin; mso-fareast-font-family:Calibri; mso-fareast-theme-font:minor-latin; mso-hansi-font-family:Calibri; mso-hansi-theme-font:minor-latin; mso-bidi-font-family:"Times New Roman"; mso-bidi-theme-font:minor-bidi; mso-fareast-language:EN-US;}div.WordSection1 {page:WordSection1;}
My new comment: The Authors response is a bit contradictory. I agree with the Authors that “cell proliferation at 72 h is also limited by culture growth variables”, but this rise another observation: if so, in the figure 1 above, where the proliferation is represented as Abs, I should have a decrease at 72 hours, which does not appear in this figure
